# The volatility of beef and lamb prices in Türkiye: The role of COVID–19, livestock imports, and energy prices

**Gurkan Bozma[1], Faruk Urak[2]\*, Abdulbaki Bilgic[3], Wojciech J. Florkowski[4]**

**1** Department of Economics, College of Economics and Administrative Sciences, Iğdır University, Iğdır, Türkiye, **2** The Turkish Radio and Television Corporation (TRT), Erzurum Directorate, Erzurum, Türkiye, **3** Department of Management Information Systems, College of Economics and Administrative Sciences, Bilecik Şeyh Edebali University, Bilecik, Türkiye, **4** Department of Agricultural and Applied Economics, University of Georgia, Griffin, Georgia, United States of America

\* farukurak.trt@gmail.com

**Data Availability Statement:** All relevant data are within the paper and its Supporting information files.

**Funding:** The authors received no specific funding for this work.

## Abstract

This study examines the volatility of beef and lamb prices in Türkiye, as food price inflation compromises the food security of low- and middle-income households. The inflation is the result of a rise in energy (gasoline) prices leading to an increase in production costs, together with a disruption of the supply chain by the COVID-19 pandemic. This study is the first to comprehensively explore the effects of multiple price series on meat prices in Türkiye. Using price records from April 2006 through February 2022, the study applies rigorous testing and selects the VAR(1)–asymmetric BEKK bivariate GARCH model for empirical analysis. The beef and lamb returns were affected by periods of livestock imports, energy prices, and the COVID-19 pandemic, but those factors influenced the short- and long–term uncertainties differently. Uncertainty was increased by the COVID–19 pandemic, but livestock imports offset some of the negative effects on meat prices. To improve price stability and assure access to beef and lamb, it is recommended that livestock farmers be supported through tax exemptions to control production costs, government assistance through the introduction of highly productive livestock breeds, and improving processing flexibility. Additionally, conducting livestock sales through the livestock exchange will create a price information source allowing stakeholders to follow price movements in a digital format and their decision-making.

## Introduction

COVID–19 started in China towards the end of 2019 and quickly became a global pandemic. The pandemic continues to have a devastating impact on many aspects of daily life, including education [1], politics [2], social relations [3], and all sectors of the economy from production to consumption [4–6]. Lockdowns [7] affected logistics, the critical part of the supply chain, while international trade decreased [8].

Consumer goods prices increased and the pandemic affected agricultural production and the food supply chain causing food prices to rapidly ascend [9]. The pandemic forced many

**Competing interests:** The authors have declared that no competing interests exist.

nations to confront the risk of food insecurity [10]. While approximately 780 million people worldwide experienced food insecurity in 2019, the number reached 927 million in 2020 as a result of the COVID–19 pandemic [11]. Food insecurity has increased rapidly during the pandemic due to the supply–demand imbalance, dramatic changes in consumer behavior, restrictions in foreign trade [12–15], and volatility in agricultural product prices [16].

Agricultural production, the origin of the food supply chain, has been one of the hardest–hit sectors during the pandemic [17, 18]. In particular, the livestock sector has been an important contributor to foreign trade, household food security, and farm income in Türkiye. The downward trend in animal production, shocked by the world COVID–19 pandemic, continued in 2021. World beef production decreased by 57.6 million tons in 2021 [19]. In Türkiye, red meat consumption per capita decreased from 6.42 kg in 2019 to 6.39 kg in 2020, while per capita meat consumption was 6.37 in 2021 kg [20]. While the nominal price of beef was around 45.64 Turkish Lira (TL)/kg in 2019, it reached 51.38 TL/kg and 62.66 TL/kg in 2020 and 2021, respectively, [21]. The nominal beef price increased by 37.3%. The pandemic–induced price increase has reduced consumer demand for animal protein.

Most of the global beef is consumed in hotels, restaurants, and other institutions [15]. The widespread closure of restaurants and catering services during the pandemic and the reduction in tourism and business travel (for example, Salazar, Bergstrom [22]) have lowered the demand for beef [23, 24]. The persistence of the pandemic strengthened the expectation that the decrease in beef production and trade would continue [25], reducing red meat consumption, an important animal protein source. Inevitably, low- and middle–income households with high shares of food expenditures risk food insecurity, especially protein consumption. This raises concerns about the impact of COVID–19 on agricultural production, which could become a major threat to the long–term food supply and food security [26].

The Global Food Security Index used to compare Türkiye's pre–and post–COVID periods, showed the food security score dropped from 63.5 to 61.2, i.e., by 2.3 points between 2019 and 2020 [27]. Moreover, the share of the population with an income under $3.20/day (2011 purchasing power parity–PPP) has increased from 1.5% in 2019 to 2.2% in 2021 in Türkiye [28]. Additionally, the contraction in production due to COVID–19 was accompanied by hoarding, increasing price volatility [23]. Food price volatility is a threat to food security and has been a major concern in Türkiye [17]. While the producer price of red meat in Türkiye in March 2020 increased by approximately 15% [21], wholesale prices increased by 28% year–to–year [29]. Retail meat prices, on the other hand, increased by 20% [30]. Before the pandemic, increases in producer prices were between 3% and 11%, wholesale meat prices increased between 2% and 12%, and retail prices were between 0.99–10% [21, 29, 30]. According to Metropoll [31], 61.8% of the respondents stated they stopped eating meat, while 50.3% emphasized they reduced their meals. Connors, Malan [32] also reported that respondents in their study stated they have been consuming less meat due to price increases caused by COVID–19.

Although most of the studies conducted in recent years have tried to link the transmission of price volatility between agricultural markets to the energy market [33–35], some researchers, albeit a small number, focused directly on the diffusion of price volatility among agricultural markets [36–40]. Among agricultural commodity markets, little is known about how instability in prices is transmitted along the food chain in Türkiye, particularly before the outbreak and during the COVID–19 pandemic, and its link to red meat imports. Also, there is a lack of studies addressing the asymmetric spread of price volatility in agricultural supply chains during the pandemic and in response to imports. It has been well established that the increase in agricultural prices does not have the same effect as the decrease in prices. The burden and benefits of the sudden price fluctuations are unevenly distributed across markets [40] as they threaten food security by deterring producers and limiting consumer from access to

adequate food volume. Therefore, this study aims to empirically elicit the pass–through of price uncertainty in Turkish red meat markets (beef carcass and lamb carcass–high protein foods) in the context of the meat and livestock import decision, the COVID–19 pandemic, and the energy price changes reflected in gasoline prices. This study reveals for the first time how the ongoing structural changes due to both import decisions and the pandemic shape (expand or contract) the red meat supply in Türkiye, both in the return markets and in the transmission of long–term price volatility using the Baba, Engle, Kraft, and Kroner (BEKK) generalized bivariate autoregressive conditional heteroscedasticity (BGARCH) model (thereafter, BEKK BGARCH). The study contributes to the existing literature in several ways. First, the study uncovers how both individual returns and the spread of individual long–term price volatility in beef and lamb markets counteracts the structural changes induced by the COVID–19 pandemic and meat imports. Second, the study shows is how the changes in the level of gasoline prices determine both returns and individual long–term price volatility spillovers in both the beef and lamb carcass supply. Since gasoline prices (energy prices), are an important part of the price formation of agricultural products, it is important to know how resilient are both the returns and volatility of beef and lamb to changes in the energy price. The earlier studies mostly focused on the effects of energy returns on agricultural commodity returns but ignored the direct effects of price.

## COVID–19 pandemic

The pandemic affects food security in four areas: availability, accessibility, utilization, and stability [41]. Disruptions in any of the four areas create food price uncertainty affecting household food insecurity and the production decisions of producers.

COVID-19 has triggered prices in terms of supply and demand, causing food security problems around the world. Insufficient access to agricultural inputs and related services, labor shortages, feed shortages in animal feeding and rising agricultural input prices are supply-side factors that cause food prices to rise and fluctuate excessively. On the other hand, problems in food logistics, increases in both retail prices and household food consumption, the decline in both consumers' purchasing power and international trade, and stockpiling effects are demand-side driving forces in the rise of food prices [41–43]. Increasing food prices adversely affect both the accessibility and sustainability of food, causing pressure on unstable (perishable) foods and nutritional preferences with decreasing income. Such consequences complicate the effective use of food, tolling deaths from COVID–19 and lasting the problem of long-term health recovery. During the COVID–19 pandemic, agricultural production experienced severe disruption of input supply. The major disruption was the decrease in oil production, which affected the supply of imported inputs used in agricultural production [44] as well as having a significant direct and indirect impact on a large variety of economic sectors [45, 46]. The bidirectional volatility spillover between energy and agricultural products has become more pronounced recently [47]. The energy sector has already felt the effects of COVID–19, mainly resulting from demand shocks [48] lowering oil prices and weakening the supply [49].

Apart from the energy supply disruption, the labor shortage caused by the pandemic also affected agricultural production. The production disruptions caused by COVID–19 are mostly concentrated in labor–intensive sectors [7, 50, 51]. A sudden restriction of mobility due to the closure of national borders has led to labor shortages in countries dependent on seasonal migrant workers in the agri–food sector [52]. Seasonal lack of migrant labor has compromised food availability, causing prices to skyrocket globally [52, 53].

The quarantine of the workforce due to COVID–19 exacerbated the labor supply problems. Labor shortages dampened agricultural returns [54]. While highly mechanized in developed

and emerging economies, harvesting in low–income countries still requires manual labor and animal traction, and labor–related problems induced or exacerbate food insecurity [55, 56]. For example, in April 2020, more than 4,900 positive COVID–19 cases were reported among approximately 500,000 employees at 115 meat processing plants in the United States resulting in the temporary closure of 40 meat processing and packaging facilities [51]. The closure reduced beef and pork supplies to retailers by an estimated 25 percent over three weeks [56]. Meat processing plant closures due to COVID–19 limited access of urban residents to meat and fish [57]. Douglas [58] reported that approximately 250 people in meatpacking companies died from COVID–19 between the beginning of April 2020 and the end of the year, when the pandemic rapidly expanded in the United States.

Decreased demand, due to the decline in purchasing power, affected the ability and willingness of farmers to invest and adopt suitable measures to ensure the supply of livestock. Farmers lost control over food production [59]. Research in low–and middle–income countries has shown that prices have been rising in food–insecure regions. For example, in Ethiopia, high prices were cited as the biggest challenge, with around 90% of households reporting not getting enough food [60], while 45% of those without access to basic foods in Mali cited rising food prices as a major problem [61, 62].

## Literature summary

The price of food is the most important factor for individual well–being [63]. Rising food prices reduce welfare and limit individual access to food. [64–66] have emphasized that high food prices cause higher general price inflation affecting the poor the most. Layani, Bakhshoo-deh [67] concluded that increases in food prices reduce the income and purchasing power of individuals living in urban areas. Access to food, constrained by price shocks, creates deprivation and poverty [68]. Ultimately, food price shocks influence household food security.

Similarly, Kirikkaleli and Darbaz [69] emphasized that food security concerns are triggered by market volatility due to limited exports and tighter price controls. It has also been claimed that oil–importing countries are strongly affected by market volatility. Price increases compromise food security limiting both diet diversity (downstream impact) and food share in consumption expenditures (upstream impact) [70].

The COVID–19–induced uncertainties in food prices have micro- and macro–level effects. The excessive increase in food prices, loss of purchasing power, and decrease in household access to food adversely affect the profitability of food manufacturing companies and their investment decisions. Uddin, Alam [71], on the other hand, stated that uncertainties in food prices have negative effects on firm performance due to COVID–19. In Türkiye, low–income families were most affected during the pandemic period [72]. Low–income families allocated smaller shares to food and health expenditures due to loss of income, compromising affordability. Overall, fluctuations in food prices during the pandemic affected farmer income and consumer purchasing power in underdeveloped and developing countries [73, 74].

In addition to the loss of income and uncertainties occurring simultaneously, the disrupted supply and production encouraged individuals to stock up on food, deepening the demand imbalance [23, 75, 76]. For example, high volatility in rice prices was observed, especially at the start of the pandemic [77]. Food security, nutrition, and labor could be impacted in the short, medium, and long term by the COVID–19 global pandemic [78]. Food supply instability and rising prices have severely reduced the poor's access to basic foodstuffs in India. Poor women and children have been especially vulnerable to COVID–19–related price increases and food shortages [79]. Ijaz, Yar [7] reported that prices for meat and meat products have risen due to the disruptions in the supply chain greatly reducing the poor's access to meat, and risking their

**Table 1. Empirical studies of effects of COVID–19 on food prices.**

| Source | Geographical focus | Methodology | Summary of results |
|---|---|---|---|
| Akter [86] | EU | Difference–in–Differences | HICPs for meat, fish, and seafood climbed considerably in March 2020 in countries with serious limitations compared to nations with relatively low or no stay–at–home restrictions. |
| Connors, Malan [32] | Case Study | – | Food has become expensive due to COVID–19 and difficult for individuals to buy food. Several participants stated that they bought less meat due to the price increases caused by COVID–19. |
| Goeb, Maredia [87] | Myanmar | Fixed Effect Regression | The increased margins were small, only 2% of head rice prices for consumers and producers alike. |
| Mahajan and Tomar [85] | India | Linear regression | Online product availability drops by an average of 10% but has a limited impact on online prices. |
| Ramsey, Goodwin [88] | USA | TAR and VECM | Considerable price fluctuations in the beef market as a result of COVID–19. |
| Ruan, Cai [89] | China | Time regression discontinuity | Vegetable price increases peaked in the fourth week of the lockdown but gradually returned to the previous level by the eleventh week. Supply chain disruptions drive price increases. |
| Wang, Shao [90] | Global | Multifractal detrended cross–correlation analysis | Following the advent of COVID–19, except for the orange juice futures market, the multi–fractal cross–correlation of all agricultural futures has risen. |
| Yan, Cai [16] | 71 countries | 2SLS | Trade distortions led to further volatility of world agricultural prices by 22% during the COVID–19 pandemic. |
| Yu, Liu [84] | China | GARCH | The COVID–19 outbreak has had a significant and negative impact on pork prices in Beijing, China. |
| Zamani, Bittmann [76] | Iran | Engle and Granger | Beef retail prices showed higher markup and asymmetries during the lockdown. |

food security. Laborde, Martin [41] also noted that the price increases caused by COVID–19 have contributed to food insecurity. Aday and Aday [80] emphasized that COVID–19 affects the entire supply chain from the farm to the consumer, emptying shelves and increasing meat prices. Income loss significantly affected meat production [81]. For example, lamb slaughter decreased by 25.9% between April 2019 and April 2020 in Spain [82]. There has been a 25% decrease in demand for chicken with the closing of restaurants due to COVID–19 in Pakistan [83]. Yu, Liu [84] and Zamani, Bittmann [76] investigated the red meat market. The response to COVID–19 included a limitation on in–person interactions. Mahajan and Tomar [85] addressed online prices during COVID–19, whereas Yan, Cai [16] stressed the effect of trade bans on the prices of agricultural products.

Despite numerous studies of COVID–19's effects on the food sector, there has been a lack of empirical studies addressing the impact of COVID–19 on the risk associated with red meat prices, i.e., beef and lamb, important sources of dietary protein. The present study fills this gap. Table 1 illustrates empirical studies that examined the effect of COVID–19 on food prices.

## Materials and methods

### Data

The beef carcass and lamb carcass prices used in the study were obtained from the daily stock market values of the Turkish Union of Chambers and Commodity Exchanges (TOBB). The gasoline pump prices, obtained from the database of the Energy Market Regulatory Authority (EMRA), were used to measure the effect of oil price volatility on beef and lamb carcass prices. Türkiye started to import live animals and carcass meat from various countries, especially after 2010, since the domestic supply could not meet the increasing demand for meat in response to the growing urban population, high immigration rate, and tourism. The live cattle imports met the demand for red meat and prices stabilized. To mark periods when livestock and

carcass meat were imported, a dummy variable was added to the empirical analysis. The import volume series was obtained from the Turkish Statistical Institute (TSI) database.

Beef and lamb carcass prices were deflated using the food price index, while gasoline prices were deflated using the energy price index. After matching the records of beef, lamb, and gasoline prices, a total of 474 observations recorded between April 2006 and February 2022 were used. This study used the robust method to correct the standard deviations of the parameters of the conditional variance–covariance model to preempt concerns about the limited number of observations to estimate the GARCH model [91]. Rezitis and Ahammad [92] estimated a multivariate GARCH model by applying a similar method to a series of 50 observations with robust correction. Moreover, Gardebroek and Hernandez [93] and Wu, Li [94] applied the multivariate GARCH model without using the robust method although the number of observations in the two sub–samples (before and after 2006) was smaller than the number of observations in the current study.

Türkiye had to implement policies that triggered carcass and livestock imports to reduce domestic prices in red meat prices and ensure the supply–demand balance [95]. Since 2017, the country has been supplying inexpensive imported meat to contracted markets through the state–supported Meat and Milk Institution [96]. In 2017, the customs duty was reduced to 26% due to the high rate of increase in red meat prices in Türkiye, causing excessive economic burden on consumers directly and indirectly through catering services [97]. Although tax reductions encouraged imports to Türkiye in 2017, price stability in the red meat market was not be achieved and the country once again reduced the customs duty on carcass imports to 40% in 2018 [95]. In addition, Agricultural Credit Cooperative Markets, which became operational in March 2017 under the Ministry of Agriculture and Forestry (TOB) to have a price–regulating authority in the food market compared to other private sector chain markets (such as A101, BİM, ŞOK, etc.), offered limited but more affordable food in chain stores, which have become widespread in the country.

### Econometric approach

First, the lag length is determined in the mean equations used to elicit the effects of both imports and the COVID–19 pandemics on the beef and lamb carcass returns and uncertainty transmissions. The lag length in the autoregressive process in the mean equation was one (1) based on the application of several information criteria (Akaike, Schwartz, Hanan–Quin). As a result, the model was defined as VAR(1)–asymmetric BEKK BGARCH. This model includes conditional vector autoregression equation and the bivariate GARCH in the conditional variable variance equation Based on the assumption that positive and negative shocks are of equal magnitude, the model is superior to other models because it adds asymmetric effects to the conditional variance equation and this feature has been preferred by researchers [38, 39, 98, 99]. The VECH model, known as the vector stacking covariance matrix, has lost its attractiveness because the covariance matrix, known as H, does not meet the semi–definite positive condition and has problems with convergence in the log–maximum likelihood function with excessive parameter loading. On the other hand, even if the constant conditional correlation (CCC) model satisfies the semi–definite positive condition of the variance–covariance matrix, it is inadequate in handling the overload of parameters and the time–independent nature of the conditional correlation. Although the DCC model, which has a time–varying (dynamic) conditional correlation due to its lower parameter load, provides the semi–definite positive condition in the variance–covariance matrix, the current study applies the BEKK model because it allows asymmetric information flow among markets [98–100].

Recent empirical studies have reported the positive and negative ebb and flow in returns, and changes in volatility have had different effects [98, 99]. Therefore, the asymmetric parameter is included in the conditional variance equation to test the presence of asymmetric effects. Eqs 1 and 2 below show the mean and conditional variance equations specified for beef and lamb carcass returns. The distinct mean equations are:

$$R_{v,t} = \alpha_{0,v} + \alpha_{1,v}R_{v,t-1} + \alpha_{2,v}\mathrm{Pr}G_{t-1} + \alpha_{3,v}R_{cv,t-1} + \alpha_{4,v}R_{cl,t-1} + \alpha_{5,v}R_{iv,t-1} + \varepsilon_{v,t}$$

$$R_{l,t} = \delta_{0,l} + \delta_{1,l}R_{l,t-1} + \delta_{2,l}\mathrm{Pr}G_{t-1} + \delta_{3,l}R_{cv,t-1} + \delta_{4,l}R_{cl,t-1} + \delta_{5,l}R_{il,t-1} + \varepsilon_{l,t} \quad (1)$$

$$\varepsilon_{j,t} = H_{j,t}^{1/2}\eta_{j,t}, j = v, l$$

where $v$ and $l$ stand for beef and lamb carcass. $R_v$ and $R_l$ show the returns on the series of two types of meat and are regressed on their lagged values. Preliminary results, such as the cross effects of lamb carcasses show the beef carcass market return equation, but the cross effects of beef carcass return in the lamb carcass market return equation were statistically insignificant, and the cross effects were omitted in each mean return equation.

Meanwhile, the term "price volatility" is used to describe the directionless price propagations of a commodity. Volatility is measured by the daily percentage difference in the price of the commodity, for example, $R_t = 100*\ln(\mathrm{Pr}_t-\mathrm{Pr}_{t-1})$, where $\mathrm{Pr}_t$ and $\mathrm{Pr}_{t-1}$ reflect the real price of the commodity in the current market and the previous period and ln stands for natural logarithm, while $R_t$ reflects the return of the commodity. Volatility is not related to the level of prices, but by the familiar definition, the degree of variation defines it and its effects are elicited by measuring the variance equation. Since price is a component of supply and demand, volatility transmission is inherently a result of these two unavoidable basic supply and demand characteristics of the market. Therefore, high or low swing levels in market prices reflect extraordinary characteristics of supply and/or demand.

The $R_{cv,t-1}$, and $R_{cl,t-1}$ variables show the one lagged beef carcass and lamb carcass returns in the COVID–19 period, respectively, while variables $R_{iv,t-1}$, and $R_{il,t-1}$ are the lagged amount of return when live cattle/beef carcass and the lagged amount of return when live sheep/lamb and sheep/lamb carcasses were imported, respectively. $\mathrm{Pr}G_{t-1}$ reflects the one–lagged real gasoline price. The current study used the real gasoline price in the meat market return series. In countries like Türkiye, with a vulnerable economy (almost all oil is imported), the prices of oil are perceived quickly by the market rather than the return. Therefore, the real price of gasoline in the return equation and the time–varying conditional variance equation are preferred. The estimated parameters of the corresponding variables in each return equation are $\alpha_{j,v}$, and $\delta_{jl}$ where $j = 0,...,5$. The constants in each return equation are $\alpha_0$ and $\delta_0$, while $\eta_{lm}$ (m = v and l) is independently and identically distributed random noise corresponding to each return series. The gasoline series return was ignored in the return equations because it was statistically insignificant.

$H_t$ matrix is defined below. The generic form of the conditional variances of the AR(1)–asymmetric BEKK BGARCH model is:

$$H_t = \Upsilon\Upsilon' + A'\varepsilon_{t-1}\varepsilon'_{t-1}A + B'H_{t-1}B + D'\xi_{t-1}\xi'_{t-1}D \quad (2)$$

The matrix H consists of two parts. The first part includes the equation constants and can be expressed as $\Upsilon = (C + \Phi\,\mathrm{Pr}G_{t-1}+ \Gamma R_{cv,t-1} + \Lambda R_{cl,t-1} + \Psi R_{iv,t-1} + \Omega R_{il,t-1})$, where C, $\Phi$, $\Gamma$, $\Lambda$, $\Psi$, and $\Omega$ are 2x2 upper triangle matrices that measure the constant coefficients of time–varying conditional variance equations, pass–through of real gasoline prices ($\mathrm{Pr}G_{t-1}$), beef and lamb carcass returns in the COVID–19 period ($R_{cv,t-1}$ and $R_{cl,t-1}$), and beef and lamb carcass returns in the import period ($R_{iv,t-1}$ and $R_{il,t-1}$). The second part of equation (2) includes

short–term shocks ($\varepsilon_{t-1}$), long–term volatility ($H_{t-1}$), and asymmetric effects ($\xi_{t-1}$), expressed as $A'\varepsilon_{t-1}\varepsilon'_{t-1}A + B'H_{t-1}B + D'\xi_{t-1}\xi'_{t-1}D$. Matrices A and B contain the parameters of short–term shocks and long–term volatility, respectively, while matrix D includes parameters expressing asymmetric transmission. In addition, diagonal coefficients in A, B, and D matrices are only related to their shocks, volatility, and asymmetry of the relevant market, while those outside the diagonal in these matrices define the presence of uncertainty transmission from one market to another. The bivariate BEKK model allows $H_t$ to have positive definite residuals ($\varepsilon_t$), softening the symmetry assumption by creating different relative responses to positive–negative shocks via the conditional variance–covariance matrix ($H_t$). The estimators of the model were obtained using the maximum likelihood technique.

## Results

### Descriptive statistics and preliminary analysis

Table 2 shows that the lamb price return was higher than the beef price return. Considering the unconditional variances obtained from standard deviations of returns of both beef and lamb, the price of lamb seems to display higher volatility than the price of beef during the studied period. As inferred from Fig 1, the synchronized relationship between the two meat types shows that both markets have common dynamics, whilst especially after 2008, the real price of lamb had almost doubled by the end of 2011. Grain prices increased after 2008 because some volume was allocated to biofuel production. Additionally, due to urban population growth and foreign immigration, Türkiye experienced high red meat prices. After 2011, sheep production increased and the larger supply caused real prices to decrease. Meanwhile, livestock imports that started in 2009 could have exerted downward pressure on prices.

During the COVID–19 periods, the agricultural product prices in Türkiye increased due to the increase in input prices (Fig 1). Major inputs used in agricultural production (fertilizer, feed, pesticides, seeds) are imported. Furthermore, the rise in the exchange rate (weakening Turkish lira) and the decrease in foreign trade affected food prices during the COVID–19 pandemics. Also, COVID–19 has negatively affected agricultural production such as meat, a source of protein, by mitigating household purchasing power [101]. Real beef prices dramatically increased as the first effects of COVID–19 occurred (Fig 1).

Gasoline has lower returns and higher volatility than beef or lamb. Interestingly, the real price of gasoline (Table 2) has a negative mean return and is the least volatile among all considered series. The observed pattern is a domestic policy result (e.g., government control) rather than an effect of price volatility of the world crude oil market.

The kurtosis coefficient indicates that the series under consideration has a leptokurtic (fat–tail) distribution and suggests that the autoregressive conditional heteroscedasticity (ARCH) effect may be present in the series. The skewness coefficient shows the probability of a positive or negative return in the return series. Beef, lamb, and gasoline series are more likely to lose, on average, in the markets. The Jarque–Bera statistics obtained using the skewness and kurtosis coefficients show that the return series is not normally distributed over time. Table 2 correlation coefficients between return variables show a positive, but very small, a correlation between gasoline and beef (0.083) returns, while the returns of lamb price and gasoline (0.012) are least correlated. A positive correlation (0.099) between the returns of the two red meat series indicates that a unit change in the standard deviation of any of the two variables resulted in a variation of about 0.099 in the other variable linear standard deviation.

Further tests probed if autocorrelation affected the series (Ljung–Box (LB–Q), McLeod–Li, and Hosking's Multivariate Q–statistics (HM–Q)). The calculated LB–Q and McLeod–Li test values for each return series and multivariate HM–Q statistics reject the null hypothesis that

**Table 2. Descriptive statistics and unit root test results.**

| Statistics | Returns | | |
|---|---|---|---|
| | $R_{v,t}$ | $R_{l,t}$ | $R_{g,t}$ |
| Mean | −0.006 | −0.005 | −0.054 |
| Std. dev. | 3.840 | 5.314 | 2.700 |
| t–statistics (mean = 0) | −0.036 (0.971) | −0.022 (0.981) | −0.435 (0.663) |
| Skewness | 0.004 | 0.108 | 1.708 |
| Kurtosis | 1.557 | 2.620 | 20.613 |
| Jarque–Bera | 46.488 *** (0.000) | 132.497 *** (0.000) | 8367.000 *** (0.000) |
| *Correlations between real price levels* | | | |
| $Pr_{v,t}$ | 1.000 | | |
| $Pr_{l,t}$ | 0.323 | 1.000 | |
| $Pr_{g,t}$ | 0.007 | −0.274 | 1.000 |
| *Correlations between return series* | | | |
| $R_{v,t}$ | 1.000 | | |
| $R_{l,t}$ | 0.099 | 1.000 | |
| $R_{g,t}$ | 0.083 | 0.012 | 1.000 |
| *Correlations between squared return series* | | | |
| $R^2_{v,t}$ | 1.000 | | |
| $R^2_{l,t}$ | 0.111 | 1.000 | |
| $R^2_{g,t}$ | −0.039 | −0.064 | 1.000 |
| *Testing serial correlation among return series of beef and lamb prices* | | | |
| LB–Q (10) | 47.123 *** (0.000) | 37.380 *** (0.000) | |
| McLeod–Li (10) | 35.340 *** (0.000) | 56.001 *** (0.000) | |
| HM–Q (10) | 111.215 *** (0.000) | | |
| *Testing ARCH effects among return series of beef and lamb prices* | | | |
| ARCH–LM (10) | 3.604 *** (0.000) | 4.493 *** (0.000) | |
| MARCH–LM (10) | 42.100 *** (0.000) | | |
| HM–Q$^2$ (10) | 107.005 *** (0.000) | | |
| *Testing unit root among beef and lamb prices* | | | |
| ADF | −17.558 *** (lags = 1) | −16.804 *** (lags = 1) | |
| KPSS | 0.019 (lags = 1) | 0.022 (lags = 1) | |

Note:

*, **, and *** are statistically significant at 10%, 5%, and 1% respectively and p–values are in parenthesis. $R_{g,t}$ stands for the return of gasoline.

the series are not autocorrelated. Results imply that increasing returns often follow increasing returns, and decreasing returns follow decreasing returns for the next reported observation [39, 102]. The tests were followed by testing for heteroskedasticity (ARCH–Lagrange multiplier, hereafter ARCH–LM), and HM–Q$^2$ and multivariate ARCH–LM hereafter MARCH–LM). The individual ARCH–LM and multivariate ARCH–LM tests detect heteroskedasticity in the price series and indicate clustering in the ARCH residues of the series. The clustering of residuals shows that increases in return series follow increases and decreases follow decreases [39, 103], supporting earlier findings [104–106].

The Augmented Dickey–Fuller (ADF) unit root test [107] and the Kwiatkowski–Phillips–Schmidt–Shin (KPSS) test determined the stationarity of the series (Table 2). The ADF and KPSS unit root test results show that the return series is stationary at a 1% significance level.

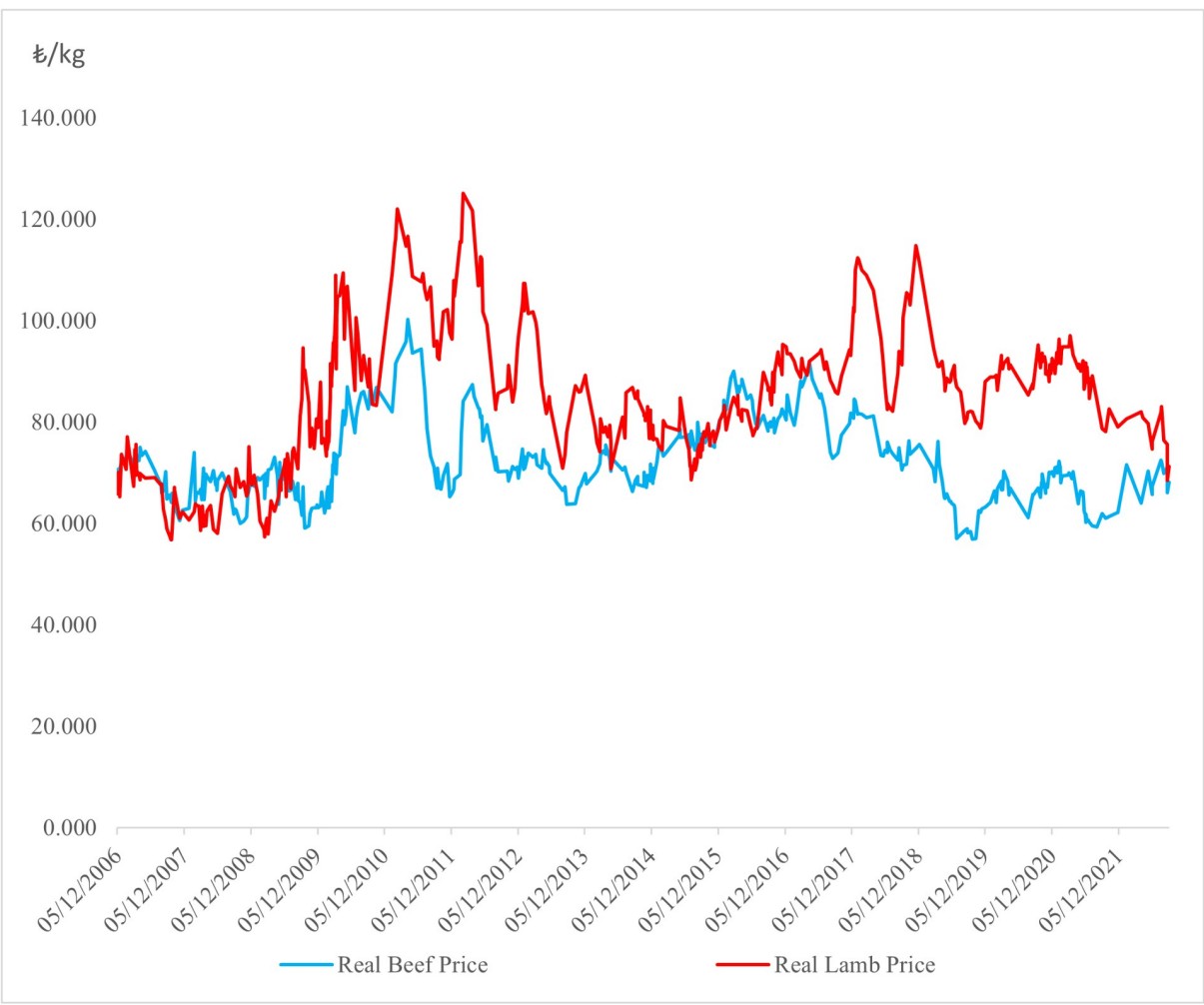

**Fig 1. Daily real prices of beef and lamb.**

### Time–varying returns to the beef and lamb series

Considering the distribution of the return series over time, high amplitudes are observed especially in the beef and lamb carcass return series after 2008 (Fig 2). A very narrow amplitude is present from 2013 to the beginning of 2018. When the returns of financial instruments begin to increase or decrease, the pattern continued on the next trading day. Especially during the COVID–19 periods, consecutive positive returns were detected on the trading days of beef and lamb carcass returns. Such a situation arises from the imbalance between supply and demand due to supply chain disruptions.

Fig 3 shows the squared returns of beef and lamb and confirms the presence of the ARCH effect in the returns. The stability of the beef returns between 2012 and 2018 is attributed to the adequate red meat supply supplemented by imported livestock, in addition to the Ministry of Agriculture and Forestry support policies to increase domestic livestock production. The steady supply reduces price volatility, preventing market speculation. COVID–19 affected logistics and trade restrictions on cattle imports, while the increase in meat demand caused sudden volatility in previously stable returns.

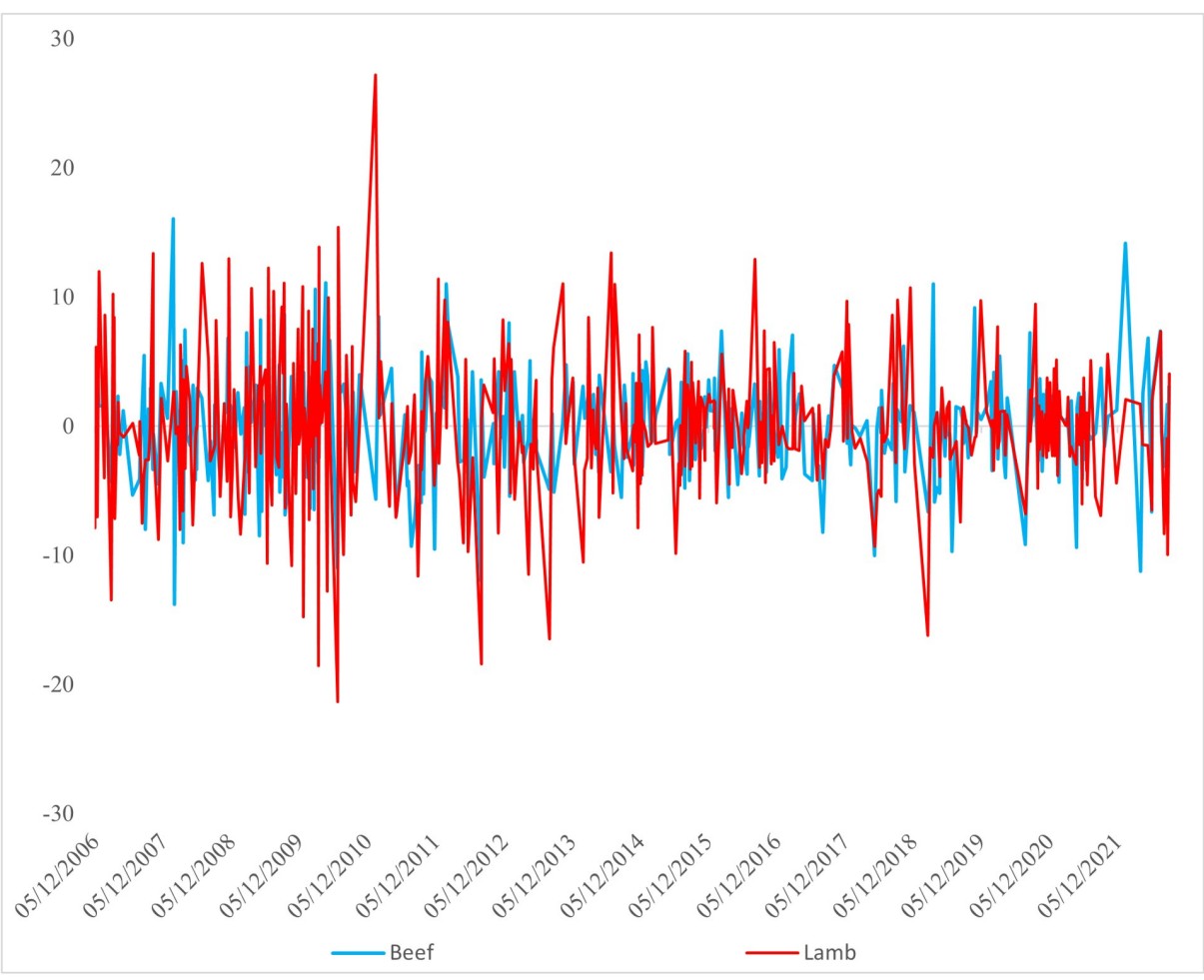

**Fig 2. Time–varying return series of beef and lamb.**

## Discussion

Table 3 shows parameter estimates from the mean and variance equations obtained using the VAR(1)–asymmetric BEKK BGARCH model for both types of red meat. The mean equations were established in different forms of an AR model than in some earlier studies [39, 102].

The beef carcass returns are affected negatively by their one–period lagged values ($\alpha_{1,v} = -0.311$). Similarly, the lamb carcass prices are negatively and significantly affected by their own lagged values ($\delta_{1,l} = -0.263$). A 1% increase in beef and lamb prices lagged one period lowering their current period returns by 0.31% and 0.26%, respectively. The effect of the past beef price is larger than the effect of lamb prices. Rezitis [108] found that the veal return was positively affected by the first and second lag values, while a negative effect was associated with the third lag value. Rezitis [108] also found that the return on poultry was negatively affected by the lagged values.

A lag in the return of gasoline has a different effect. While the increase in gasoline return increased the return on beef, it decreased the return on a lamb. Beef production uses more fossil fuels than lamb [109]. It appears that the increases in gasoline prices would be reflected in the beef prices, further increasing beef returns over lamb. The imports of meat and livestock mitigate the returns on beef and lamb. A decrease in the domestic red meat supply causes food

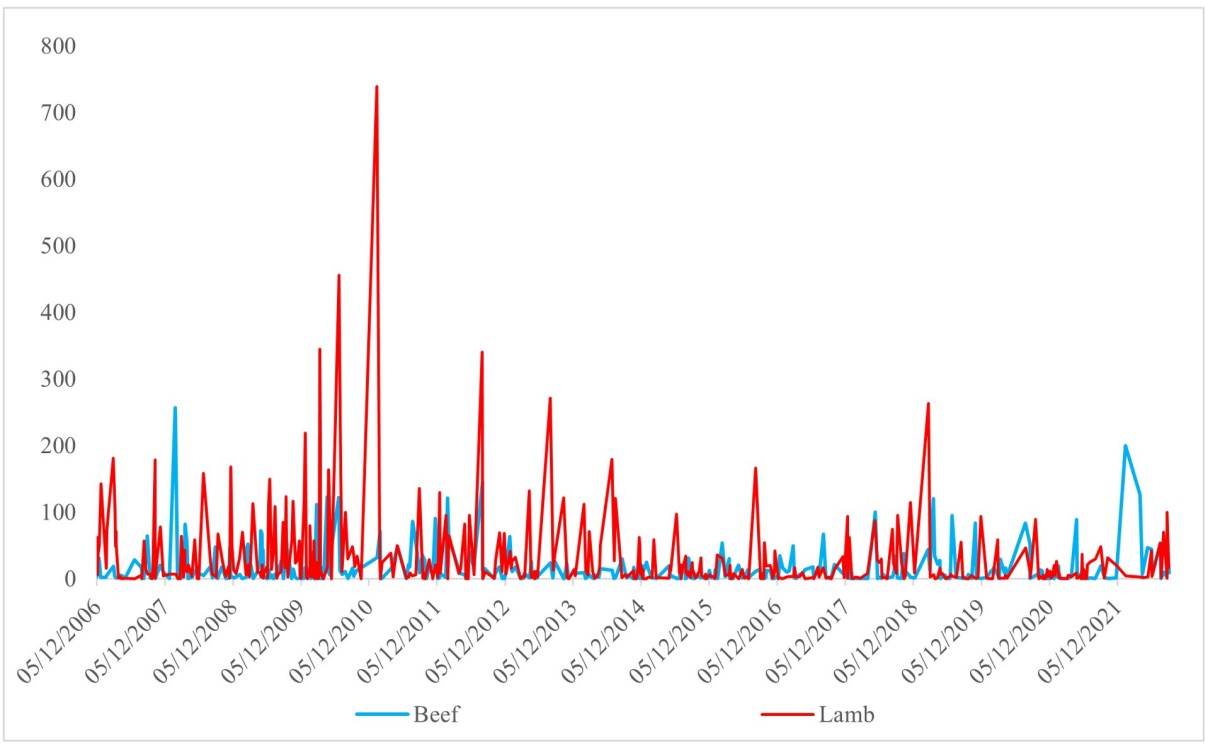

**Fig 3. Squares of time–varying return series of beef and lamb.**

price increases and contributes to inflation affecting consumers. Live animal importation is supported by the government to reduce the effects of food insecurity caused by food price increases and reduced accessibility. Thus, from 2010 to 2011, the supply of red meat increased by approximately 20% [110]. The government reduced the customs duty on imported livestock from 135% to 26% and on carcass meat from 100–225% to 40%, [111]. Imports increased the number of live cattle number by 7% from 2017 to 2018. Also, while the price of lamb increased by 14.78% in 2017, it decreased by 14.48% in 2018. The annual increase in the price of beef was reduced from 9.62% in 2017 to 4.78% in 2018 [21]. The red meat import policy had a positive effect on food security through reduced price and increased supply on the domestic market. Although food security has been positively affected, some farmers raising livestock could face losses due to low prices.

The import tax reductions are consistent with a monetary policy to ease short–run food inflation, but there still is a legitimate debate as to whether low–and middle–income families will be able to purchase an adequate amount of food [112]. Although the red meat producers in the country are harmed by the import policy, the policy is important to ensure continuous supply and food inflation affecting consumers. One of the causes of the supply disruption is large number of intermediaries and the lack of competition in the food sector in Türkiye [113]. The margin added at every stage of the supply chain causes consumer prices to increase while the share of producers shrinks.

The increase in oil prices in the lagged period increases the beef carcass return ($\alpha_{2,v}$ = 0.060) while decreasing the lamb carcass return ($\delta_{2,l}$ = −0.203). While the beef carcass increase reflects energy costs, adding to inflation, the prices of grazing lambs decrease during the fattening period. Increasing energy prices directly affect beef prices. However, the reverse situation in lamb carcass prices reflects how little energy is used by raising sheep.

**Table 3. Parameter estimates for both mean returns and conditional variances.**

| Parameters | Returns | |
|---|---|---|
| | $R_{v,t}$ | $R_{l,t}$ |
| *Mean return estimates:* | | |
| Constant | −0.819 * (0.443) | 4.315 *** (1.166) |
| $R_{v,t−1}$ | −0.311 *** (0.044) | – |
| $R_{l,t−1}$ | – | −0.263 *** (0.045) |
| $PrG_{t−1}$ | 0.060 ** (0.025) | −0.203 *** (0.070) |
| $R_{cv,t−1}$ | 0.098 ** (0.040) | 0.147 * (0.078) |
| $R_{cl,t−1}$ | −0.070 ** (0.028) | −0.113 *** (0.056) |
| $R_{iv,t−1}$ | −0.002 (0.004) | – |
| $R_{il,t−1}$ | – | −0.015 ** (0.006) |
| *Conditional variance estimates:* | | |
| *Constants in $\Upsilon'\Upsilon$:* | | |
| $c_{1i}$ | −0.437 (0.707) | |
| $c_{2i}$ | 0.203 (1.037) | 0.133 (1.315) |
| *Parameters associated with gasoline price ($PrG_{t−1}$):* | | |
| $\Phi_{1i}$ | 0.123 ** (0.060) | – |
| $\Phi_{2i}$ | 0.147 * (0.082) | 0.073 (0.062) |
| *Parameters associated with beef price return in the COVID-19 period ($R_{cv,t−1}$):* | | |
| $\Gamma_{1i}$ | 0.226 *** (0.062) | – |
| $\Gamma_{2i}$ | 0.337 *** (0.058) | −0.131 *** (0.030) |
| *Parameters associated with lamb price return in the COVID-19 period ($R_{cl,t−1}$):* | | |
| $\Lambda_{1i}$ | −0.175 *** (0.047) | – |
| $\Lambda_{2i}$ | −0.260 *** (0.044) | 0.090 *** (0.022) |
| *Parameters associated with beef price return in the import period ($R_{iv,t−1}$):* | | |
| $\Psi_{1i}$ | −0.028 ** (0.012) | – |
| $\Psi_{2i}$ | −0.041 *** (0.013) | −0.010 (0.013) |
| *Parameters associated with lamb price return in the import period ($R_{il,t−1}$):* | | |
| $\Omega_{1i}$ | 0.031 ** (0.013) | – |
| $\Omega_{2i}$ | 0.026 (0.017) | 0.009 (0.012) |
| *ARCH parameters:* | | |
| $a_{1i}$ | 0.361 ** (0.161) | 0.217 (0.142) |
| $a_{2i}$ | −0.134 ** (0.058) | −0.121 (0.128) |
| *GARCH parameters:* | | |
| $b_{1i}$ | 0.762 *** (0.256) | −0.124 (0.239) |
| $b_{2i}$ | −0.108 * (0.057) | 0.870 *** (0.064) |
| *GARCH asymmetric parameters:* | | |
| $d_{1i}$ | −0.041 (0.241) | −0.040 (0.117) |
| $d_{2i}$ | −0.0009 (0.106) | 0.241 ** (0.098) |

Note: Standard errors are in parentheses.

*, **, and *** are statistically significant at 10%, 5%, and 1% respectively

## Effects of COVID–19 on beef and lamb returns

The examination of the effects of COVID–19 shows that it had distinct effects on beef and lamb returns. COVID–19 had a positive and statistically significant effect on beef ($\alpha_{1,v}$ = 0.098) and lamb ($\alpha_{3,v}$ = 0.098) carcass returns, while lamb carcass returns during COVID–19

($\alpha_{4,v} = 0.147$) were negative and statistically significant with response to beef ($\delta_{3,l} = -0.070$) and lamb ($\delta_{4,l} = -0.113$) carcass returns. Türkiye was affected by the closure of international boundaries due to COVID–19 restricting trade and logistics. According to Comtrade [114], live cattle imports decreased by 23.32% by weight in the first quarter of 2020 compared to the same quarter in 2019, 63.75% in the second quarter, 66.38% in the third quarter, and 6.04% in the fourth quarter. Sheep imports increased by 3.26% in the first quarter of 2020 but decreased by 36.41% and 22.79% in the second and third quarters, respectively. The beef carcass imports also declined. COVID–19 caused a decrease in livestock and carcass imports, lowering the supply of red meat and risking food insecurity. The current study shows, as expected, the beef carcass return decreased during the periods of imports ($\alpha_{5,v} = -0.002$). The lamb carcass returns also declined ($\delta_{5,l} = -0.015$), but the effect on the beef carcass returns is statistically insignificant.

## Short–term shocks and long–term volatility

In terms of uncertainty, the market returns of beef are directly affected by short–term shocks ($a_{11} = 0.361$) and long–term volatility ($b_{11} = 0.762$) (Table 3). Specifically, one unit of uncertainty in their short–term shocks increases the long–term persistent uncertainty in the beef price return by 0.36 units, while the unit change in volatility in the previous period increases the uncertainty in the market return of beef by 0.76 units. The interpretation of news by the market as good or bad, which affects the instability of beef prices in the short term (ARCH effect), will affect prices but by less in the long run (GARCH effect). Therefore, increased uncertainty in the short and long run will increase volatility. Mitigating risk creates more uncertainty, which disrupts domestic production and rises the risk of food insecurity.

The structure of the meat market contributes to long–term persistent instability besides the current beef price instability. Such relationships characterize the current unstable situation of the red meat market in Türkiye. The red meat market in Türkiye is complex. Increases in input costs (feed, oil, electricity, etc.) and the animal food pricing in the market contribute to red meat price volatility. Between 2005 and 2012, food prices increased by 93% in Türkiye. The substantial increase in retail beef and lamb prices by 106% and 138%, respectively, contributed to a surge of general food inflation and inevitably affected low–and middle–income families [112, 115]. Similarly, retail prices of milk increased by 39% [115], while the animal feed index increased by 100% in the 2005–2012 period. According to Akgunduz, Cilasun [113], there is a link between milk and beef prices. The lagging milk price causes cows to be slaughtered and increases beef supply, lowering prices in the short–term, which later resume their upward movement. During the outbreak of COVID–19, milk prices increased by 20% in April 2020 year–to–year. The price increase in May 2020 was 18% year–to–year [21]. The stable relationship between red meat and milk deteriorated as the COVID–19 pandemic started, reducing animal slaughter, and the milk–beef price relationship favored milk. It is estimated that red meat production decreased by 6.6% in 2020 compared to 2019 [116]. Chung and Myers Jr [117] and Gustafson [68] emphasized that price increases affect food security for the poor.

The volatility of beef price returns is negatively affected by both short–term shocks ($a_{21} = -0.134$) and long–term volatility ($b_{21} = -0.108$) of lamb price returns. The long–term volatility (GARCH effect) that dominates the lamb carcass market exposed that market's unprotected position ($b_{22} = 0.870$). Considering long–term uncertainty, the lamb carcass market is more vulnerable to volatility than the beef carcass market ($b_{11} = 0.76$), which would worsen if the lamb carcass supply is disrupted. Long–term supply–side factors (GARCH causes) such as imports, diseases, low fertility, farmers withdrawing from production, etc. could lead to

excessive lamb return volatility. Interestingly, in terms of long–term uncertainties in the markets, the pass–through of asymmetric information was statistically significant only in the lamb carcass market ($d_{22}$ = 0.241) and the impacts of negative news differed greatly from that of positive news, which is likely to primarily affect the lamb carcass market. Short–term negative shocks in the lamb carcass market have a substantial impact on long–term supply uncertainty by discouraging farmers' long–term production decisions. The result coincides with results from the vector error correction (VEC) model [115] that price transmission for beef is asymmetric in Türkiye. Ozertan et al. (2015) found that wholesale prices in the red meat market adjust faster than retail prices and the relative price stickiness at the retail level increases as the price margins increase. The main reason for the existence of asymmetric price pass–through in the red meat sector, an indication of oligopolistic market behavior, is the supermarket chain market power and persistent imperfect retail competition. Earlier, Rezitis and Stavropoulos [118] and Luo and Liu [119] established the existence of asymmetrical volatility spillovers in pork, beef, mutton, and chicken prices.

## Gasoline price effects

Increases in gasoline prices directly ($\Phi_{11}$ = 0.123 and $\Phi_{22}$ = 0.073) and indirectly ($\Phi_{21}$ = 0.147) increase long–term beef carcass return volatility. Such outcomes are likely due to high production costs forcing farmers to choose alternative products by shortening the beef supply chain. Earlier studies identified the volatility of world crude oil prices to influence prices of a wide range of agricultural products [106, 120–122]. Moreover, Wu and Li [123] found an asymmetric effect of world crude oil prices on the agricultural products market, while Nazlioglu, Erdem [124] did not confirm such volatility spillover before the food crisis of 2006–2008. The latter study also confirmed a unidirectional movement from the wheat market to the world crude oil price.

In the post–crisis period, Nazlioglu, Erdem [124] confirmed a dual (bi–directional) spillover between crude oil and agricultural product markets, especially from the wheat market to the world crude oil market. Similar findings were also reported by Gardebroek and Hernandez [93]. Chen, Kuo [125] and Rezitis [126] showed a close relationship between oil prices and the prices of agricultural products. Roman, Górecka [127] found that there is a long–term relationship between oil prices and meat prices among several agricultural products they examined. Finally, Akgunduz, Cilasun [113] reported that the increase in barley (used in biofuel production) prices had negative effects on the livestock sector. The findings of the current study coincide with the previously reported findings on agricultural and energy prices [106, 120–125, 128–133].

## COVID–19 effects

Table 3 also shows the effects of COVID–19 on the return volatility of beef and lamb carcasses are statistically significant at the 1% level. The variable specified as a product of the COVID–19 dummy and the beef carcass market return increases the latter volatility ($\Gamma_{11}$ = 0.226) but reduces the volatility of the lamb carcass market return ($\Gamma_{22}$ = –0.131). Additionally, the beef carcass returns in the COVID–19 period reinforced the long–term uncertainty in the beef carcass return over the lamb carcass market ($\Gamma_{21}$ = 0.337). However, the effect of variability in lamb carcass return during the COVID–19 period was found to be the opposite of the previous finding. Although it makes the long–term risk in its market permanent ($\Lambda_{22}$ = 0.090), it has had a direct ($\Lambda_{11}$ = –0.175) and an indirect ($\Lambda_{21}$ = –0.260) healing effect on the beef carcass market.

During the pandemic, food prices increased due to the decrease in supply and the increase in household saving habits. Connors, Malan [32] emphasized that the price increases caused by COVID–19 changed household consumption habits, and they bought less meat. Hobbs [23] stated that after the COVID–19 outbreaks, households increased savings due to anxiety as prices increased, causing food insecurity concerns. Kirikkaleli and Darbaz [69] suggested that limiting exports also triggered market volatility. The COVID–19–induced slowdown in international trade caused the upward price movement [16]. The swings in agricultural commodity prices increase producer risk exposure and affect producers' current and future investment decisions [120, 121] while threatening consumer food security, especially in countries where the household food expenditure share of total expenditures is large [120, 121, 134, 135].

## Imports effects

While import decisions may provide consumers with access to relatively inexpensive meat, imports may also result in loses to farmers through low prices, causing red meat prices to rise in the long run due to the short domestic supply. Changes in the lamb carcass return during the periods of imports are statistically significant only in the beef market ($\Omega_{11} = 0.031$). The changes in the beef carcass return during periods of imports reduce the direct ($\Psi_{11} = -0.028$) and indirect ($\Psi_{21} = -0.041$) long–term uncertainty of the beef market. Such findings contradict the finding reported by Chadwick and Baştan [112], who stated that the import decisions reduce temporarily the uncertainty in the cattle market in Türkiye. Considering long–term disruptions in the red meat supply and the impact of high energy costs, food inflation could be inevitable in Türkiye.

## Beef and lamb conditional variances and correlations

Fig 4 shows how the conditional variances of the beef and lamb carcass returns fluctuate in the period under study. The reason for the excessive variability in the conditional variances of beef

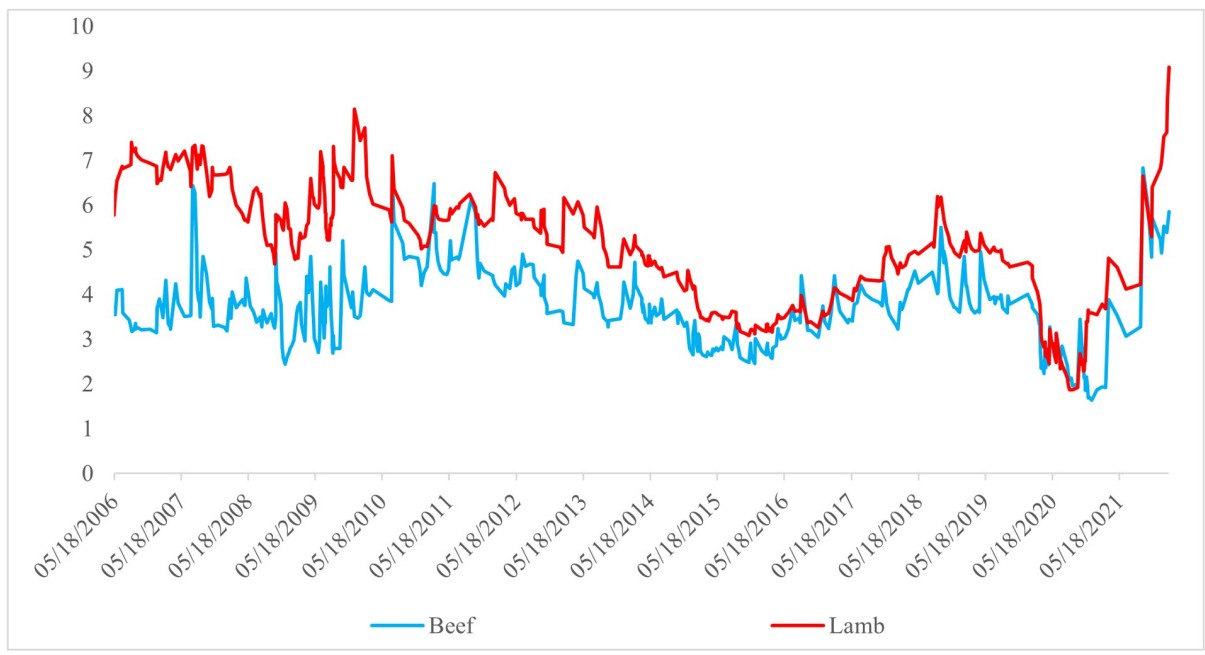

**Fig 4. Time–varying conditional variances of beef and lamb.**

and lamb carcasses between 2008 and 2010 was the diversion of grains from feed to biofuel production. The diversion increased uncertainty pass–through to the prices of agricultural products and the crude oil market. The above–average increases in feed prices increased the uncertainty in the red meat market. Although the policy of importing carcass meat after 2010 stabilized the meat market, the global COVID–19 pandemic increased the prices of agricultural products since early 2020. The conditional volatility of beef and lamb carcasses peaked in June 2020 (Fig 4). Since beef and lamb carcasses are substitutes, the distribution of uncertainties is synchronously similar over time.

Fig 5 depicts the conditional correlation between beef and lamb, confirming the time–varying uncertainty relationship between the two types of meat in Türkiye. The average conditional correlation coefficient between beef and lamb carcass of 0.16 indicates that the uncertainty in either market triggers a positive simultaneous change in the other market. While the conditional correlation between beef and lamb carcass uncertainties was negative once COVID–19 affected Türkiye, the conditional correlation effect was positive in mid–2020 when the effect of the pandemic was fully felt. Next, the effect declined to the extremely negative and only peaked later. Such unpredictable swings in the two markets highlight the uneasiness of farmer production decisions and the rise of food insecurity in households.

LB–Q, McLeod–Li, ARCH–LM, and multivariate tests (HM–Q and MARCH–LM) verified whether individual error terms include autocorrelation and time–varying variance problems (Table 4). Error terms obtained from each model were tested with LB–Q, McLeod–Li, and multivariate HM–Q tests at 10 lag lengths. The results of the tests confirmed the absence of autocorrelation. Also, the individual ARCH–LM and multivariate ARCH–LM tests did not detect heteroskedasticity in error terms. Finally, the tests concluded that the VAR(1)–asymmetric BEKK BGARCH model was valid for the two meat price series.

Granger causality test results in panel B of Table 4 were obtained from the mean and variance equations. First, the alternative hypothesis of whether gasoline prices, beef carcass, and

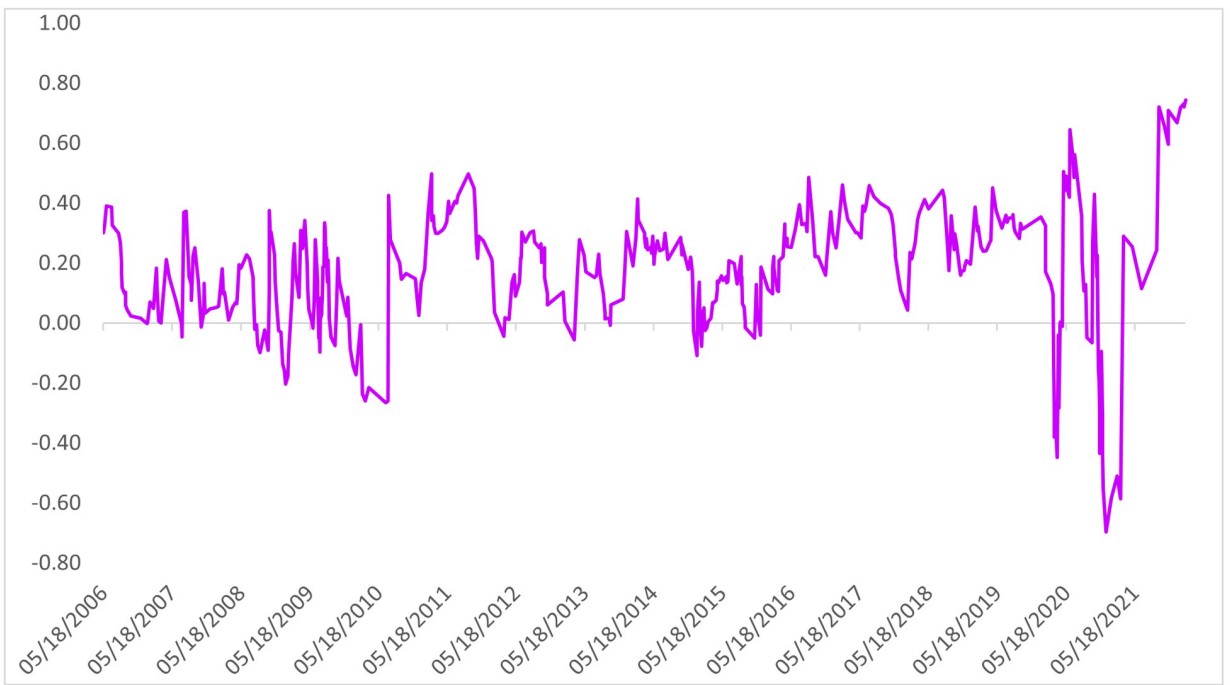

**Fig 5. Time–varying conditional correlation between beef and lamb.**

**Table 4. Parameter estimates for conditional variances in VAR–Asymmetric BEKK BGARCH.**

| *Panel A*: Residual Diagnostic Tests | $R_{v,t}$ | $R_{l,t}$ |
|---|---|---|
| Ljung–Box Q (10) | 13.312 * (0.091) | 15.488 (0.115) |
| *McLeod–Li (10)* | 4.456 (0.924) | 9.449 (0.490) |
| ARCH (10) | 0.426 (0.933) | 0.919 (0.515) |
| HM–Q (10) | 33.867 * (0.087) | |
| *MARCH–LM (10)* | 98.120 (0.262) | |
| *Panel B*: Model Specification Tests | | |
| Granger Causality Tests | | |
| Gasoline price, COVID–19 beef, COVID–19 lamb, and beef imports do not Granger cause beef return | | 7.197 (0.125) |
| Gasoline price, COVID–19 beef, COVID–19 lamb, and lamb imports do not Granger cause lamb return | | 115.400 *** (0.000) |
| No GARCH | $H_0$, $a_{ij} = b_{ij} = d_{ij} = 0$ for all i, j = 1,2,3 | 5388.405 *** (0.000) |
| Diagonal GARCH | H0: All off–diagonal elements of A, B, and D are jointly zero | 21.587 *** (0.001) |
| No Asymmetry | $H_0$, $d_{ij} = 0$ for all i, j = 1,2,3 | 11.605 ** (0.020) |
| Off–diagonal gasoline prices in the conditional variance equations are jointly zero | | 10.470 ** (0.014) |
| Off–diagonal of COVID–19 beef in the conditional variance equations are jointly zero | | 39.579 *** (0.000) |
| Off–diagonal of COVID–19 lamb in the conditional variance equations are jointly zero | | 53.562 *** (0.000) |
| Off–diagonal of import beef in the conditional variance equations are jointly zero | | 10.687 ** (0.013) |
| Off–diagonal import lamb in the conditional variance equations are jointly zero | | 8.725 ** (0.033) |

Note:

*, **, and *** indicate the significance at 10%, 5%, and 1%, respectively; probability in parentheses.

lamb carcass returns during COVID–19 and imports of live lamb and beef carcasses affected beef carcass returns was rejected (Table 4). Second, a simultaneous causality relationship was determined by accepting the alternative hypothesis that gasoline prices, beef carcass, and lamb carcass returns during the period of the pandemic, and livestock import decisions affect lamb carcass returns. The hypothesis of the absence of the GARCH relationship that all parameters of short–term shocks, long–term volatility, and asymmetric effects in the conditional variance equation are simultaneously zero was rejected. The result shows that the return uncertainties of beef and lamb carcasses and their conditional variances are determined by uncertainties of news from the market.

The rejection of the null hypothesis (Table 4) regarding the absence of asymmetric effects supports such influence. The null hypotheses showing that gasoline returns do not Granger cause the uncertainty of beef and lamb carcasses were rejected. The fuel price increase may leave the farmer undecided on whether to produce. Dixit and Pindyck [136] emphasized that uncertainty hurts investment decisions. Pinno and Serletis [137] showed that oil price uncertainty reduces food production. Also, Mensi, Hammoudeh [106], Nazlioglu, Erdem [124], Hanson, Robinson [130], Mitchell [132], Nazlioglu and Soytas [133] found that crude oil price affects agricultural production and agricultural commodity prices.

Inflation can induce food insecurity by reducing household purchasing power. Gumus, Olgun [138] found that low–and middle–income families did not consume an adequate amount of meat. The current study established that COVID–19 had a statistically significant effect (1%) on the beef and lamb carcass return uncertainty. Baffes and Wu [9], Yan, Cai [16], Qingbin, Liu [18], Connors, Malan [32], Ramsey, Goodwin [88] stated that COVID–19 hurt

agricultural production and the prices of agricultural products. Ramsey, Goodwin [88] emphasized that meat prices have increased in the United States due to COVID–19 and reported that surveyed consumers admitted they bought less meat as a result of the price increase caused by COVID–19 [32]. The null hypothesis stating that the decision to import livestock and carcass meat does not Granger cause the uncertainty of beef and lamb returns was rejected (at the 5% significance level), showing that import decisions curbed uncertainty and suppressed the rising red meat prices in Türkiye.

## Conclusion

The global COVID–19 pandemics deeply affected economic, social, and cultural relationships. The ravages of the pandemic are related to the size of a country's economy, but it affected all countries regardless of economic development level. The agricultural sector, which secures basic nutrition, has affected farmers and left consumers to grapple with high inflation, including staple foods. The red meat sector, a source of animal protein essential in healthy diets, has been also severely affected. The current study empirically demonstrated how volatility in beef or lamb carcass prices affects each other in the context of energy prices (specifically gasoline prices), livestock imports, and the COVID–19 pandemics in Türkiye. The elicited effects of gasoline price, imports, and COVID–19 on the returns in two red meat markets and the uncertainty shifts provide insights for making decisions fitting the disruptions that undermine the food supply in Türkiye.

Results indicate that the time–varying conditional variances of beef and lamb carcass returns are affected by both short–term shocks and long–term volatility. The beef and lamb price uncertainties are affected differently by positive and negative news in the market. The negative news emerging from the red meat market and the wrong direction of markets are a result of anti–competitive actions and may deepen the red meat market uncertainty and negatively affect producer decisions. Otherwise, increased volatility may affect retail prices and consumers, especially those from low–income households which may become food insecure. The red meat market faced increased uncertainty during the COVID–19 pandemics, resulting in runaway food inflation, including meat prices. The annual inflation rate is currently around 70% in Türkiye.

The eventful periods such as pandemics can cause major disruptions in the food supply having vital consequences for consumers and producers. To ensure price stability during periods of economic instability, policymakers should have knowledge of the critical markets such as agricultural commodity markets and, especially, the meat markets which proved susceptible to events such as the pandemic. Our findings imply that essential government initiatives, especially subsidy programs, are required to protect producers during the period of big price declines, and to shield consumers from sudden high prices of staples. Policy recommendations to lessen the price uncertainty caused by COVID–19 on the red meat market are outlined. First, it is vital to increase support to small and medium–sized farms producing livestock. This policy will encourage production expansion and ensure a sustainable supply of red meat. Farmers can be granted tax–exemptions to alleviate input costs (primarily fuel). Similarly, to mitigate increasing production costs, the government can provide machinery–equipment grants to support farmers producing animal feed. Second, to rid the country of unproductive livestock breeds, farmers can qualify for breed–specific grants to assure high productivity. The promotion of high–yielding breeds is a priority of the Ministry of Agriculture and Forestry (MAF). Considering that there are many stakeholders such as traders, brokers, shippers, and retailers in the food distribution sector in Türkiye, producer associations should be allowed to offer butcher services to increase the farmer share of the value generated along the supply

chain. Supporting local stock markets and local food markets facilitating direct interaction between farmers and consumers will reduce supply uncertainty, as well as temper price increases and reduce food insecurity. In addition, by conducting all transactions of livestock sales through the livestock exchange, a price information source will be created allowing all stakeholders to easily follow price movements in a digital environment. Our empirical results confirm that the possible long–term persistent uncertainties resulting from bottlenecks in the beef carcass supply chain can be alleviated through imports, incentivizing efficiency improvements in the domestic red meat supply chain.

## Supporting information

**S1 Table. Nominal price of commodities.**
(XLSX)

**S2 Table. Real price of commodities.**
(XLSX)

**S1 Data.**
(RAR)

## Acknowledgments

The authors thank the referees for their constructive suggestions and comments, whose comments were constructive in improving the manuscript. The authors also express their appreciation to Laura Alfonso, John Cruickshank, and Tydaisha White for assistance in the preparation of the manuscript.

## Author Contributions

**Conceptualization:** Gurkan Bozma, Faruk Urak, Abdulbaki Bilgic, Wojciech J. Florkowski.

**Data curation:** Gurkan Bozma, Faruk Urak.

**Formal analysis:** Abdulbaki Bilgic, Wojciech J. Florkowski.

**Funding acquisition:** Wojciech J. Florkowski.

**Investigation:** Gurkan Bozma, Faruk Urak, Abdulbaki Bilgic.

**Methodology:** Gurkan Bozma, Faruk Urak, Abdulbaki Bilgic.

**Supervision:** Abdulbaki Bilgic, Wojciech J. Florkowski.

**Writing – review & editing:** Gurkan Bozma, Faruk Urak, Abdulbaki Bilgic, Wojciech J. Florkowski.

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
