## [Decision Letter · Decision Letter 0]

19 Dec 2022

PONE-D-22-29212The volatility of beef and lamb prices in Turkey: The role of COVID-19, livestock imports, and energy pricesPLOS ONE

Dear Dr. Urak,

Thank you for submitting your manuscript to PLOS ONE. After careful consideration, we feel that it has merit but does not fully meet PLOS ONE’s publication criteria as it currently stands. Therefore, we invite you to submit a revised version of the manuscript that addresses the points raised during the review process.

Specifically, the paper was assessed by three reviewers, and they all agreed that the results are not appropriately presented.The revised paper should follow PLOS ONE's format including referencing style.  Also, the language of the paper needs proofreading. Thus, it is recommended to get the paper professionally proofread by a specialized English proofreader.

We look forward to receiving your revised manuscript.

Kind regards,

Mohammed Al-Mahish

Academic Editor

PLOS ONE

Journal Requirements:

"No funding resources"

4. We note that Figure 1 in your submission contain copyrighted images. All PLOS content is published under the Creative Commons Attribution License (CC BY 4.0), which means that the manuscript, images, and Supporting Information files will be freely available online, and any third party is permitted to access, download, copy, distribute, and use these materials in any way, even commercially, with proper attribution. For more information, see our copyright guidelines: http://journals.plos.org/plosone/s/licenses-and-copyright.

Reviewers' comments:

Reviewer's Responses to Questions

**Comments to the Author**

1. Is the manuscript technically sound, and do the data support the conclusions?

Reviewer #1: Yes

Reviewer #2: Partly

Reviewer #3: Yes

2. Has the statistical analysis been performed appropriately and rigorously? 

Reviewer #1: Yes

Reviewer #2: Yes

Reviewer #3: Yes

3. Have the authors made all data underlying the findings in their manuscript fully available?

Reviewer #1: No

Reviewer #2: Yes

Reviewer #3: Yes

4. Is the manuscript presented in an intelligible fashion and written in standard English?

Reviewer #1: Yes

Reviewer #2: No

Reviewer #3: Yes

5. Review Comments to the Author

Reviewer #1: The volatility of beef and lamb prices in Turkey: The role of COVID-19, livestock imports, and energy prices

Summary:

The paper addresses the impact of three variables (Covid-19, livestock import, and Energy prices) on beef and lamb prices in Turkey. Using data covering the period of April 2006 through February 2022, the study employs a VAR(1) asymmetric BEKK GARCH model for empirical analysis. The study finds different effects of different variables and offers some policy recommendations.

The paper addresses an important issue with an appropriate method. The results are interesting with sound discussion in the context of previous literature. The policy recommendations are relevant. The paper is well-structured, well-written, and well-organized.

Major issue

1. Some part of the method section is not self-explanatory and lacks some relevant information, which makes it hard for the reader to follow along. The reader is left alone in that regard. For example, it is not clear how the Covid period was singled out from the data period of 2006 and 2022.How was the COVID-19-induced uncertainties incorporated into the model? Was a dummy variable used(like the import variable, which is mentioned in this section)? If so, the method section does not indicate it. I could understand it after seeing the results in table 3. This should be clarified in the method section.

Minor

1. Line 39, “food safety” should be replaced by “food security”

2. Line 228, What do E and I denote for?

3. The official name of the country changed from “Turkey” to “Turkiye”. The new name could be used.

Reviewer #2: • The study presents an original research.

• The organization of the manuscript lacks integrity. Authors should revise the article accordingly.

• Abstract should include less details. The policy implications should be limited within the scope of key findings. The language should be carefully reviewed for clear statements.

• It is hard to comprehend the research objectives in the article. In lines (54-56) they stated that “The present study examines the pass-through of price uncertainty in Turkish red meat markets (beef carcass and lamb carcass-high protein foods), gasoline (energy) market, and in the context of meat and livestock import decisions during COVID-19”. The authors must clearly state their objectives. What is the pass-through of price uncertainty and livestock import decisions? How the authors measured it? How they defined and measured the uncertainty, returns are confusing.

• The authors should provide more evidence that their paper contributes the existing literature. In lines (56-60), the authors states “The study is first in examining the link between red meat import decisions, effects of the gasoline market, and Covid-19 on the volatility of red meat markets in Turkey using the Baba, Engle, Kraft, and Kroner (BEKK) generalized autoregressive conditional heteroscedasticity (GARCH) model (thereafter, BEKK-GARCH)”. They suggest that using an existing econometric method (BEKK-GARCH) is a novelty. But they did not provide enough evidence about the compatibility, importance, and superiority of this specific method.

• Despite studying a specific country, manuscript lacks a brief discussion on the governmental policies for the sector in question. Particularly, the policy implications of the article recommend the government for taxation and subsidies.

• The authors should use more understandable abbreviations for variables in the tables.

• Conclusions are partly presented in an appropriate fashion and are supported by the data.

• The article is poorly presented in an intelligible fashion and is written in standard English.

• The manuscript has grammatical errors and unnecessary wordings. Several grammatical errors must be corrected and the authors must avoid unnecessary wordings. The language of the manuscript must be carefully revised in order to have a clear understanding.

• The research meets all applicable standards for the ethics of experimentation and research integrity.

• The article adheres to appropriate reporting guidelines and community standards for data availability.

Reviewer #3: The authors provide a well-stated and well-performed analysis. However, I have some concerns:

Caution is needed for the correlation results. If data do not follow the normal distribution instead of Pearson correlation, Spearman correlation should be employed.

In general, authors should be more analytical in presenting their results.

6. PLOS authors have the option to publish the peer review history of their article (what does this mean?). If published, this will include your full peer review and any attached files.

Reviewer #1: No

Reviewer #2: No

Reviewer #3: **Yes: **Anthony N. Rezitis

---

## [Author Response · Author response to Decision Letter 0]

7 Feb 2023

Reviewer #1: The volatility of beef and lamb prices in Turkey: The role of COVID-19, livestock imports, and energy prices

Summary:

The paper addresses the impact of three variables (Covid-19, livestock import, and Energy prices) on beef and lamb prices in Turkey. Using data covering the period of April 2006 through February 2022, the study employs a VAR(1) asymmetric BEKK GARCH model for empirical analysis. The study finds different effects of different variables and offers some policy recommendations.

The paper addresses an important issue with an appropriate method. The results are interesting with sound discussion in the context of previous literature. The policy recommendations are relevant. The paper is well-structured, well-written, and well-organized.

Answer: Thank you for the favorable comments.

Major issue

1. Some part of the method section is not self-explanatory and lacks some relevant information, which makes it hard for the reader to follow along. The reader is left alone in that regard. For example, it is not clear how the Covid period was singled out from the data period of 2006 and 2022. How were the COVID-19-induced uncertainties incorporated into the model? Was a dummy variable used(like the import variable, which is mentioned in this section)? If so, the method section does not indicate it. I could understand it after seeing the results in table 3. This should be clarified in the method section.

Minor

1. Line 39, “food safety” should be replaced by “food security”

Answer: Thanks. It has been corrected.

2. Line 228, What do E and I denote for?

Answer: Thank you for your careful reading. The omission has been fixed.

3. The official name of the country changed from “Turkey” to “Turkiye”. The new name could be used.

Answer: It is true, as the United Nations declared Turkey as Türkiye. Thank you for the correction.

Reviewer #2: • The study presents an original research.

• The organization of the manuscript lacks integrity. Authors should revise the article accordingly.

Answer: The manuscript has been revised and we hope meets the expectations. Thank you for the comment.

• Abstract should include less details. The policy implications should be limited within the scope of key findings. The language should be carefully reviewed for clear statements.

Answer: The abstract was rewritten following your comment.

• It is hard to comprehend the research objectives in the article. In lines (54-56) they stated that “The present study examines the pass-through of price uncertainty in Turkish red meat markets (beef carcass and lamb carcass-high protein foods), gasoline (energy) market, and in the context of meat and livestock import decisions during COVID-19”. The authors must clearly state their objectives. What is the pass-through of price uncertainty and livestock import decisions? How the authors measured it? How they defined and measured the uncertainty, returns are confusing.

Answer: We are grateful to the Referee for bringing those issues to our attention. At the Referee’s request, we rewrote the introduction by making the objectives straightforward and more understandable – see the text highlighted in yellow. Also, information about the price volatility has been detailed in footnote #1 in the Materials and Methods section. 

• The authors should provide more evidence that their paper contributes the existing literature. In lines (56-60), the authors states “The study is first in examining the link between red meat import decisions, effects of the gasoline market, and Covid-19 on the volatility of red meat markets in Turkey using the Baba, Engle, Kraft, and Kroner (BEKK) generalized autoregressive conditional heteroscedasticity (GARCH) model (thereafter, BEKK-GARCH)”. They suggest that using an existing econometric method (BEKK-GARCH) is a novelty. But they did not provide enough evidence about the compatibility, importance, and superiority of this specific method.

Answer: We are grateful for the comments and have updated the relevant section (Introduction) as requested, and listed the paper’s contribution. The revised text now includes an explanation of the superiority of the GARCH method in the Materials and Methods section. The added text is highlighted in yellow.

• Despite studying a specific country, manuscript lacks a brief discussion on the governmental policies for the sector in question. Particularly, the policy implications of the article recommend the government for taxation and subsidies.

Answer: We are grateful for the comment and have updated the relevant section. Please see the data section and the text colored in yellow.

• The authors should use more understandable abbreviations for variables in the tables.

Answer: Thank you -the abbreviations and tables have been rechecked and corrected.

• Conclusions are partly presented in an appropriate fashion and are supported by the data.

Answer: We are grateful for this comment. We have revised the relevant section at the request of the referee (see the text highlighted in yellow).

• The article is poorly presented in an intelligible fashion and is written in standard English.

Answer: The revised paper was edited by a professional English editor. We thank you for the comment.

• The manuscript has grammatical errors and unnecessary wordings. Several grammatical errors must be corrected and the authors must avoid unnecessary wordings. The language of the manuscript must be carefully revised in order to have a clear understanding.

Answer: The revised paper was edited by a professional English editor.

• The research meets all applicable standards for the ethics of experimentation and research integrity.

Answer: We thank you.

• The article adheres to appropriate reporting guidelines and community standards for data availability.

Answer: We thank you.

Reviewer #3: The authors provide a well-stated and well-performed analysis. However, I have some concerns:

Caution is needed for the correlation results. If data do not follow the normal distribution instead of Pearson correlation, Spearman correlation should be employed.

Answer: We are grateful to the Referee for bringing the issue to our attention; the necessary corrections have been made and included in footnote 2 as thanks to the referee.

In general, authors should be more analytical in presenting their results.

Answer: All presented results were reviewed, and integrity was ensured in the text.

---

## [Editor Report · Decision Letter 1]

13 Feb 2023

PONE-D-22-29212R1The volatility of beef and lamb prices in Türkiye: The role of COVID–19, livestock imports, and energy pricesPLOS ONE

Dear Dr. Urak,

Thank you for submitting your manuscript to PLOS ONE. After careful consideration, we feel that it has merit but does not fully meet PLOS ONE’s publication criteria as it currently stands. Therefore, we invite you to submit a revised version of the manuscript that addresses the points raised during the review process.

ACADEMIC EDITOR Comments:Please revise the manuscript according to PLOS ONE's formatting and referencing style. The references are not written in accordance with the journal's style, and you can use a referencing software such as Mendely to help you formatting your references in accordance with PLOS ONE's style. Also, the clean copy of the paper should be free from highlights.

We look forward to receiving your revised manuscript.

Kind regards,

Mohammed Al-Mahish

Academic Editor

PLOS ONE
---

## [Author Response · Author response to Decision Letter 1]

14 Feb 2023

Reviewer #1: The volatility of beef and lamb prices in Turkey: The role of COVID-19, livestock imports, and energy prices

Summary:

The paper addresses the impact of three variables (Covid-19, livestock import, and Energy prices) on beef and lamb prices in Turkey. Using data covering the period of April 2006 through February 2022, the study employs a VAR(1) asymmetric BEKK GARCH model for empirical analysis. The study finds different effects of different variables and offers some policy recommendations.

The paper addresses an important issue with an appropriate method. The results are interesting with sound discussion in the context of previous literature. The policy recommendations are relevant. The paper is well-structured, well-written, and well-organized.

Answer: Thank you for the favorable comments.

Major issue

1. Some part of the method section is not self-explanatory and lacks some relevant information, which makes it hard for the reader to follow along. The reader is left alone in that regard. For example, it is not clear how the Covid period was singled out from the data period of 2006 and 2022. How were the COVID-19-induced uncertainties incorporated into the model? Was a dummy variable used(like the import variable, which is mentioned in this section)? If so, the method section does not indicate it. I could understand it after seeing the results in table 3. This should be clarified in the method section.

Minor

1. Line 39, “food safety” should be replaced by “food security”

Answer: Thanks. It has been corrected.

2. Line 228, What do E and I denote for?

Answer: Thank you for your careful reading. The omission has been fixed.

3. The official name of the country changed from “Turkey” to “Turkiye”. The new name could be used.

Answer: It is true, as the United Nations declared Turkey as Türkiye. Thank you for the correction.

Reviewer #2: • The study presents an original research.

• The organization of the manuscript lacks integrity. Authors should revise the article accordingly.

Answer: The manuscript has been revised and we hope meets the expectations. Thank you for the comment.

• Abstract should include less details. The policy implications should be limited within the scope of key findings. The language should be carefully reviewed for clear statements.

Answer: The abstract was rewritten following your comment.

• It is hard to comprehend the research objectives in the article. In lines (54-56) they stated that “The present study examines the pass-through of price uncertainty in Turkish red meat markets (beef carcass and lamb carcass-high protein foods), gasoline (energy) market, and in the context of meat and livestock import decisions during COVID-19”. The authors must clearly state their objectives. What is the pass-through of price uncertainty and livestock import decisions? How the authors measured it? How they defined and measured the uncertainty, returns are confusing.

Answer: We are grateful to the Referee for bringing those issues to our attention. At the Referee’s request, we rewrote the introduction by making the objectives straightforward and more understandable – see the text highlighted in yellow. Also, information about the price volatility has been detailed in footnote #1 in the Materials and Methods section. 

• The authors should provide more evidence that their paper contributes the existing literature. In lines (56-60), the authors states “The study is first in examining the link between red meat import decisions, effects of the gasoline market, and Covid-19 on the volatility of red meat markets in Turkey using the Baba, Engle, Kraft, and Kroner (BEKK) generalized autoregressive conditional heteroscedasticity (GARCH) model (thereafter, BEKK-GARCH)”. They suggest that using an existing econometric method (BEKK-GARCH) is a novelty. But they did not provide enough evidence about the compatibility, importance, and superiority of this specific method.

Answer: We are grateful for the comments and have updated the relevant section (Introduction) as requested, and listed the paper’s contribution. The revised text now includes an explanation of the superiority of the GARCH method in the Materials and Methods section. The added text is highlighted in yellow.

• Despite studying a specific country, manuscript lacks a brief discussion on the governmental policies for the sector in question. Particularly, the policy implications of the article recommend the government for taxation and subsidies.

Answer: We are grateful for the comment and have updated the relevant section. Please see the data section and the text colored in yellow.

• The authors should use more understandable abbreviations for variables in the tables.

Answer: Thank you -the abbreviations and tables have been rechecked and corrected.

• Conclusions are partly presented in an appropriate fashion and are supported by the data.

Answer: We are grateful for this comment. We have revised the relevant section at the request of the referee (see the text highlighted in yellow).

• The article is poorly presented in an intelligible fashion and is written in standard English.

Answer: The revised paper was edited by a professional English editor. We thank you for the comment.

• The manuscript has grammatical errors and unnecessary wordings. Several grammatical errors must be corrected and the authors must avoid unnecessary wordings. The language of the manuscript must be carefully revised in order to have a clear understanding.

Answer: The revised paper was edited by a professional English editor.

• The research meets all applicable standards for the ethics of experimentation and research integrity.

Answer: We thank you.

• The article adheres to appropriate reporting guidelines and community standards for data availability.

Answer: We thank you.

Reviewer #3: The authors provide a well-stated and well-performed analysis. However, I have some concerns:

Caution is needed for the correlation results. If data do not follow the normal distribution instead of Pearson correlation, Spearman correlation should be employed.

Answer: We are grateful to the Referee for bringing the issue to our attention; the necessary corrections have been made and included in footnote 2 as thanks to the referee.

In general, authors should be more analytical in presenting their results.

Answer: All presented results were reviewed, and integrity was ensured in the text.

---

## [Editor Report · Decision Letter 2]

20 Feb 2023

The volatility of beef and lamb prices in Türkiye: The role of COVID–19, livestock imports, and energy prices

PONE-D-22-29212R2

Dear Dr. Urak,

We’re pleased to inform you that your manuscript has been judged scientifically suitable for publication and will be formally accepted for publication once it meets all outstanding technical requirements.

Kind regards,

Mohammed Al-Mahish

Academic Editor

PLOS ONE
---

## [Editor Report · Acceptance letter]

3 Mar 2023

PONE-D-22-29212R2 

The volatility of beef and lamb prices in Türkiye: The role of COVID–19, livestock imports, and energy prices 

Dear Dr. Urak:

I'm pleased to inform you that your manuscript has been deemed suitable for publication in PLOS ONE. Congratulations! Your manuscript is now with our production department. 

Kind regards, 

on behalf of

Dr. Mohammed Al-Mahish 

Academic Editor

PLOS ONE